# Flavonoid Components, Distribution, and Biological Activities in *Taxus*: A review

**DOI:** 10.3390/molecules28041713

**Published:** 2023-02-10

**Authors:** Qiang Wei, Qi-Zhao Li, Rui-Lin Wang

**Affiliations:** School of Medicine, Anhui Xinhua University, 555 Wangjiang West Road, Hefei 230088, China

**Keywords:** *Taxus* plants, flavonoids, structure, biological activities

## Abstract

*Taxus*, also known as “gold in plants” because of the famous agents with emphases on Taxol and Docetaxel, is a genus of the family Taxaceae, distributed almost around the world. The plants hold an important place in traditional medicine in China, and its products are used for treating treat dysuria, swelling and pain, diabetes, and irregular menstruation in women. In order to make a further study and better application of *Taxus* plants for the future, cited references from between 1958 and 2022 were collected from the Web of Science, the China National Knowledge Internet (CNKI), SciFinder, and Google Scholar, and the chemical structures, distribution, and bioactivity of flavonoids identified from *Taxus* samples were summed up in the research. So far, 59 flavonoids in total with different skeletons were identified from *Taxus* plants, presenting special characteristics of compound distribution. These compounds have been reported to display significant antibacterial, antiaging, anti-Alzheimer’s, antidiabetes, anticancer, antidepressant, antileishmaniasis, anti-inflammatory, antinociceptive and antiallergic, antivirus, antilipase, neuronal protective, and hepatic-protective activities, as well as promotion of melanogenesis. Flavonoids represent a good example of the utilization of the *Taxus* species. In the future, further pharmacological and clinical experiments for flavonoids could be accomplished to promote the preparation of relative drugs.

## 1. Introduction

The genus *Taxus* has a wide distribution throughout the world with 24 species and 55 varieties [1], in addition to nearly 400 taxoids with various skeletons having anticancer functions that have been identified [2]. Moreover, other numerous compounds with pharmacological and biological activities also accumulate in *Taxus* plants, such as flavonoids. Flavonoids originate from the Latin word “flavus” meaning “yellow” and consist of a 15-carbon skeleton, presenting a C6-C3-C6 structure consisting of two benzene rings (A and B) and one heterocyclic ring C [3].

Flavonoids are categorized into several branches based on the ring B position linkage to ring C, the unsaturated degree, and the degree of ring C openings and oxidation [3]. Until now, more and more studies on the chemical compounds of the *Taxus* genus have been performed, and over 500 compounds of plants have been reported [4], covering some subclasses such as flavones, biflavones, flavonols, dihydroflavones, dihydroflavonols, flavanols, and chalcones as shown in Figure 1.

Flavones, as a subclass of flavonoids, are characterized by a 2-phenylchromen-4-one or 2-phenyl-1-benzopyran-4-one skeleton, depicting a ring B linkage to C-2 and a double bond of C-2 to C-3 but oxygenation deficiency at C-3. Biflavones are dimers of monomeric flavonoids usually formed by attaching two flavones between C-5′ and C-8″. Flavonols have a hydroxyl substituent at C-3 of ring C compared to flavone structure, namely 3-hydroxy-2-phenylchromen-4-one. They have a distinct name from flavanols (flavonols’ first “o” is replaced with “a”) as other flavonoids such as catechin. The exclusive structural difference between flavonols and dihydroflavonols is the 2,3 double bond in flavonols. Flavanols, as derivatives of the flavans, demonstrate a 2-phenyl-3,4-dihydro-2H-chromen-3-ol skeleton, as with catechin. Chalcones are characterized by an open central heterocyclic ring C and 1,3-diphenyl-2-propenone as their core structure [3].

Nowadays, high and positive attention for health care has been focused on flavonoids. A European nutritional investigation proved that cancer had a direct relevance to flavonoid intake in people of different ethnicities [5]. Meanwhile, computer-aided workflows were used to design a drug based on flavonoid structures for a better synthesis path, bioactive effects, and few side effects to the human body [6]. Accordingly, in view of the complexity of flavonoid structure classification and diversity of the *Taxus* species, the present review aims at investigating and summarizing the rule of chemical compositions, characteristics of compound distribution, and pharmacological activities of flavonoids and analyzing the relationship between structure and activity for future research and development of the *Taxus* plants.

## 2. Chemical Structure

Growing studies on the chemical compounds of the *Taxus* genus have been exerted, and the target has been focused on 13 species (*T. fuana, T. yunnanensis, T. baccata, T. celebica, T. chinensis, T. cuspidata, T. media, T. brevifolia, T. canadensis, T. wallichiana, T. mairei, T. cuspidate var. nana, and T. chinensis var. mairei*). In all, 59 flavonoids have been identified in the *Taxus* genus, which is categorized into two flavones, 15 biflavones, four flavonols, nine flavonol glycosides, six dihydroflavones, two dihydroflavonols, one dihydroflavonol glycoside, nine flavanols, six biflavanols, and five chalcones. As far as the isolated plant parts are concerned, all these flavonoids are mainly from the needles, fruit, twigs, and leaves of *Taxus* plants, as shown in Table 1.

## 3. Results

### 3.1. Chemical Components

#### 3.1.1. Flavones

A total of two flavones **1**, **2** from *Taxus* plants have been reported as can be seen in Table 1 and Figure 2, with similar flavone skeletons, while luteolin **2** has an additional hydroxy group of C-3′ compared to apigenin **1**.

#### 3.1.2. Biflavones

Biflavones, as a group of secondary metabolites providing chemotaxonomic markers, are mainly generated from Gymnospermae including the Taxaceae and Ginkgoaceaefamilies and are formed through the phenol-oxidative coupling of flavones [70]. A total of 15 biflavones **3**–**17** have been isolated from *Taxus* plants, consisting of two apigenins or two apigenins with methyl ether, methyl, or hydroxyl groups as shown in Table 1 and Figure 2. Despite being common natural products, biflavones still exhibit special characteristics, which are shown below: Methyl ether or hydroxyl groups are attached to C-7, C-4′, C-7″, or C-4′′′, such as in compounds **3**–**6** (bilobetin, 4′′′-O-methyl amentoflavone, sciadopitysin, and ginkgetin), **8**–**13** (sequoiaflavone, isoginkgetin, putraflavone, sotetsuflavone, kayaflavone, and 4′, 7,7″-tri-O-methyl amentoflavone), and **15** and **16** (4′, 7″-di-O-methyl amentoflavone and 4″-O-methyl ginkgetin). C-7, C-4′, C-7″, and C-4′′′ are connected to a hydroxyl group in compound **7** (amentoflavone); and amethyl ether in compound **14** (4′,4″, 7,7″-tetra-O-methyl amentoflavone). C-7and C-4′′′ are connected to a methyl group, and C-5, C-4′, C-5″, C-7″, and C-3′′′ are connected to a hydroxy, such as in compound **17** (3″-hydroxy-4″, 7-dimethyl amentoflavone).

#### 3.1.3. Flavonols or Flavonol Glycosides

A coexisting double bond of C-2 to C-3 in the Cring makes flavonols and flavones become similar structures. Flavonols, as compared to flavones, present a hydroxyl substituent at the C-3 position [71]. A total of four flavonols, including **18** (kaempferol), **21** (myricetin), **23** (quercetin), and **26** (isorhamnetin) from the *Taxus* genus show a common characteristic for the C-3 hydroxy group, and nine flavonol glycosides including **19** (kaempferol-3-O-rutinoside), **20** (kaempferol-7-O-glucoside), **22** (myricetin-3-O-rutinoside), **24** (quercetin-3-O-rutinoside), **25** (quercetin-7-O-glucoside), **27** (quercetin-3-O-α-L-arabinopyranosyl-(1′′′→6″)-β-D-glucopyranoside), **28** (tricin-3-O-glucoside), **29** (quercetin-3-glucoside), and **30** (quercetin3-rhamnoside) from *Taxus* connecting to glucose, rhamnose, arabinose, or rutinose have been found in *Taxus* plants, as shown in Table 1 and Figure 2. Interestingly, there are some consistent one-to-one matched relationships between flavonol glycosides and their aglycones (flavonols), such as the flavonol glycoside of compounds **19** and **20** and their common aglycone of **18**; that of compound **22** and its aglycone **21**; and those of compounds **24**, **25**, **27**, **29**, and **30** and their common aglycone of **23**. To date, only compounds **26** and **28** have not been matched with compounds.

#### 3.1.4. Dihydroflavones, Dihydroflavonols, and Dihydroflavonol Glycosides

There are six dihydroflavones, two dihydroflavonols, and one dihydroflavonol glycoside that have been found in *Taxus* plants, which indicates the common characteristic of the saturated 2,3 double bond with two hydrogen atoms. As far as the six dihydroflavones are concerned, pinocembrin **31** can be seen as the fundamental skeleton for compounds **32**–**36** (eriodictyol, butin, naringenin, pinostrobin, and dihydrotricetin), such as in compound **32,** with its two additional hydroxyl groups at the C-4′ and C-5′ positions; compound **34,** with its additional C-4′ hydroxyl group; compound **36**, with its additional C-3′, C-4′, and C-5′ hydroxyl groups; and compounds **33** and **35**, with different special substituent groups for the C-5 and C-7 positions. At the same time, taxifolin **37** and aromadendrin **38**, as the two dihydroflavonols, have similar structures, only reflecting a hydroxy change at C5′. Aromadendrin-3-O-rutinoside **39**, as a dihydroflavonol glycoside, has **38** as its aglycone and a rutinose.

#### 3.1.5. Flavanols and Biflavanols

Flavanols possess a discriminating structural feature in that they have no oxygen-containing groups of C-4 position in the ring C, which is in common with anthocyanidins. Furthermore, the hydrogenation of at the C-2,3 double bond and a hydroxyl group linkage to the C-3 position generate two chiral centers [71]. There is a total of nine flavanols **40**–**48** that have been isolated from *Taxus* plants, which are classified into two kinds of flavanols according to different C-3 and C-4 substituent groups. As far as compounds **40**–**42** (5-deoxyleucopelargonidin, leucopelargonidin, and leucocyanidin) are concerned, they indicate the common characteristic of a 3,4-dihydroxyl group at the C-3 and C-4 positions under conditions not considering three-dimensional structure. In terms of compounds **43**–**48** ((+)-catechin, (-)-epicatechin, gallocatechin, epigallocatechin, (+)-catechin pentaaacetate, and (-)-epicatechin pentaacetate), these have three-dimensional structures with a 3-hydroxyl or acetyl oxygen group and a 3-hydrogen.

Biflavanols form a new fundamental skeleton by attaching two flavanols with linkage between C-4 and C-8″. A total of six biflavanols **49**–**54** (procyanidin B2, procyanidin B-2 decaacetate, procyanidin B-3-decaacetate, procyanidin B-4-decaacetate, afzelechin-(4α→8)-afzelechin, and afzelechin-(4α→8)-afzelechin octaacetate) present a chiral carbon at the C-3 position based on the different C-3 and C-3″ substituent groups, including hydrogen, hydroxy, and acetyl oxygen groups.

#### 3.1.6. Chalcones

A total of five chalcones **55**–**59** (pinocembrin chalcone, isoliquiritigenin, butein, homoeriodictyol chalcone, and naringenin chalcone) from *Taxus* plants present the fundamental skeleton of α, β-unsaturated ketones (*trans*-1, 3-diaryl-2-propen-1-ones), including benzene rings A and B connected to an α, β-unsaturated carbonyl group [72]. Compared with the structure of flavones **1**, **2**, they have no oxygen atom at the C-1 position and no disconnection between the oxygen atom and carbon atom at the C-2 position, but they have the same double bond at the C2/C3 position.

### 3.2. Flavonoid Properties, Extraction, and Isolation

#### 3.2.1. Physico-Chemical Properties of Flavonoids

In terms of flavonoid solubility, it is closely related to the principle of polar similarity and intermiscibility, mostly representing hydrophilic solubility of flavonoid glycosides, whereas flavonoid aglycones have lipophilic solubility. In fact, the different solubility also should link to the factors of structural alkylation, hydroxylated degree, molecule, or ion [73,74]. In addition, pH, π-conjugated system, and hydration could be seen as vital factors to color production, in which C ring saturation level and substitution of the rings A and B affect π-conjugated formation of flavonoids [75]. The color of flavonoids presents diversity characteristics, such as usually yellow in flavones, flavonols, chalcones, and aurones; red in anthocyanidins with acidic media; blue in anthocyanidins with alkaline media; colorless in catechins, flavans, and isoflavones. In addition, flavonoids can exhibit yellow or yellow fluorescence induced by UV radiation [76,77]. Special taste occurs in the different flavonoids, reflecting bitter and astringent in some flavanone glycosides such as naringin [78] while the shortage of the pyranone leads to the sweetness being magnified 1000 times in naringin dihydrochalcone compared to sucrose [79].

#### 3.2.2. Extraction and Isolation Methods

Green extraction containing the methods of ultrasound, microwave, supercritical fluid, additive enzyme, matrix solid-phase dispersion, pulsed electric field, solid-state fermentation, pressurized liquid extraction, etc. has become a new trend of extracting the flavonoid compositions, because they avoid too many solvents, time consumption, and energy cost and follow a purification process compared to the traditional methods such as Soxhlet, maceration, and boiling [80,81,82,83]. Many technical advantages of the aforementioned green extraction methods are tapped gradually, for example, based on the ultrasonic cavitation effect to break the cell wall and electromagnetic microwave with high frequency to disrupt the plant cells [81,84], which accelerate the solvent permeation and diffusion and benefit the dissolution of active components. Supercritical carbon dioxide has the characteristics of nontoxicity, high efficiency, and low temperature, which are also fit for flavonoid extraction [80]. High pressure and subcritical state in pressurized liquid extraction demonstrate the significant advantages [85].

Modern chromatographic technology is still the main technical means of flavonoid separation, showing the different traits [86]. Column chromatography, as a traditional separation method, was used to isolate quercetin **23**, morin-3-O-lyxoside from *Psidium guajava* [87]. High-performance liquid chromatography was characterized by an efficient, fast, and sensitive method, which was well fitted to the isolation of epicatchin and epgallocatechin from *Kombucha* tea [88]. Five flavonoids were separated completely from *Oxytropis falcata* Bunge by high-speed counter-current chromatography, because the method did not need a solid support and avoided sample loss, denaturation, and contamination [89]. Luteolin and apigenin were isolated from *Helichrysum chasmolycium* P.H Davis by preparative thin-layer chromatography as a fast and inexpensive method [90].

### 3.3. Flavonoid Distribution

In our study, as far as the plant parts of *Taxus* species were concerned, flavonoids mainly originated from the twigs, fruit, roots, leaves, heartwood, needles, branches, and bark of *Taxus* plants in Table 1.

In terms of the distribution of a certain flavone in *Taxus* plants, there was a special trait of the genus distribution. If biflavones were discussed separately, biflavones **5** (sciadopitysin), **6** (ginkgetin), **7** (amentoflavone), and **8** (sequoiaflavone) could be seen as the main representative compounds in the *Taxus* species because biflavones **3** and **4** (bilobetin, 4′′′-O-methyl amentoflavone) only unexpectedly occurred in the minor plants such as *T. baccata* and *T. celebica*, or were absent, such as in *T. media*. Similarly, biflavones **5** (sciadopitysin) and **7** (amentoflavone) were the dominant compounds in *T. baccata*, *T. media,* and *T. celebica*. Biflavone **6** (ginkgetin) combined the aforementioned biflavones **5** and **7** as the main components in *T. baccata* and *T. media*. A high-pressure liquid chromatographic (HPLC) analysis indicated that biflavone **7** (amentoflavone) had a high level of accumulation in *T. cuspidata,* as well as a low accumulation amount in *T. media* [14]. -Chalcone **55** (pinocembrin chalcone) was depicted as high level of accumulation in *T. mairei,* and chalcone **59** (naringenin chalcone) existed in great amounts in *T. media* and *T. cuspidata*, whereas chalcones **56**–**58** (isoliquiritigenin, butein, and homoeriodictyol chalcone) were apparently accumulated in *T. media*. -Dihydroflavone **31** (pinocembrin) was a dominant compound in *T. mairei*, while dihydroflavones **32** (eriodictyol) and **33** (butin) significantly dominated in *T. media*, and dihydroflavones **34**–**36** (naringenin, pinostrobin, and dihydrotricetin) existed at high amounts in *T. media* and *T. cuspidata*. Flavanols **40** (5-deoxyleucopelargonidin) and **41** (leucopelargonidin) were rich in *T. media*. However, flavanol **42** (leucocyanidin) was dominant in *T. mairei,* and even flavanols **43** ((+)-catechin)and **44** ((-)-epicatechin) were present in greater amounts in *T. yunnanensis* than in *T. fuana* [7,53].

Other distribution examples of different flavones were given from the literature [6], such as flavones **1** (apigenin) and **2** (luteolin), biflavones **6** (ginkgetin) and **7** (amentoflavone), and flavonols **18** (kaempferol) and **23** (quercetin), which were more predominantly accumulated in *T. fuana* than in *T. yunnanensis*. Similarly, biflavone **7** (amentoflavone), flavonol **23** (quercetin), and flavone **2** (luteolin) showed higher contents in *T. mairei* than in *T. media* and *T. cuspidata* [53].

These results indicate a distinctive value for developing and utilizing flavonoids in different *Taxus* plants because flavonoid composition is heavily influenced by environmental conditions, such as soil and climate [91,92], and the enrichment of plant composition shows complicated and inconstant variable characteristics that are greatly influenced by the genetic variants and environmental factors [93,94]. Thus, our results regarding this variation may also derive from differentiated climate conditions and ecological environments of *Taxus* plants [95]. Furthermore, more reports about flavonoid distribution have presented a certain rule, namely, that the biosynthesis of flavonoids in *Taxus* plants can be strictly limited due to biflavone distribution: biflavone dominates only in small amounts of plants, while non-dimeric flavonoids appear as dominant compounds; on the contrary, other flavonoids also exist significantly in small amounts and even accumulate as traces only when biflavonoids are mainly represented in *Taxus* plants [12,14,96,97].

### 3.4. Flavonoid Bioactivities

Flavonoids in *Taxus* species are recognized as natural bioactive components that possess wide bioactive effects, which was shown in Table 1.

#### 3.4.1. Antibacterial Activities

Four biflavones including sciadopitysin (**5**), ginkgetin (**6**), amentoflavone (**7**), and 7-O-methyl amentoflavone (**9**) from *T. baccata* presented significant antifungal activities, in which bilobetin **3** inhibited *Alternaria alternata*-, *Cladosporium oxysporum*-, and *Fusarium culmorum*—fungi with ED_50_ (median effective dose) values of 14, 11, and 17 μmol/L, respectively, and thoroughly inhibited the cultivation of *C. oxysporum* and *F. culmorum* at a concentration of 100 μmol/L. Ginkgetin **6** and 7-O-methyl amentoflavone (sequoiaflavone) **8** had stronger inhibition toward *A. alternata* at concentrations 100 μmol/L, and sciadopitysin **5** exhibited a strong inhibitory effect on *C. oxysporum* at ED_50_ value of 9 μmol/L. The bioactive results of biflavones can be related to methoxyl groups with increased or decreased antifungal activity, such as biflavones without a methoxyl group such as amentoflavone **7**, which showed inactive or weak activity against *C. oxysporum* and *A. alternata* [12]. In addition, *Neisseria gonorrhoeae,* as a kind of pathogenic, drug-resistant bacteria, was inhibited moderately by pinocembrin chalcone **55** at 128 g/mL [67].

#### 3.4.2. Antioxidant and Antiaging Activities

As is known, phenolic compounds’ radical-scavenging activity is affected by the conformational changes, substituent locations, entire amount of hydroxyl groups, and their mutual arrangement due to a molecule’s influence on metal ion chelation and sequestration [98]. DPPH (1,1-diphenyl-2-picrylhydrazyl) tests have shown that (+)-catechin **43** and (-)-epicatechin **44** at 0.01 mol/L concentrations presented high quenching results for stable radicals because both compounds possess two hydroxy groups at ortho in the benzene ring, which were found to have the strongest activities in this regard. In addition, (+)-catechin **43** and (-)-epicatechin **44** at the concentration of 0.01 mol/L isolated from *T. cuspidata* also exhibited a better inhibitory effect on the auto-oxidation for linethol than ionol, which was similar to α-tocopherol’s capability of scavenging hydroperoxide radicals and terminating chain reactions in all stages of oxidation [60]. Similarly, (+)-catechin **43** and (-)-epicatechin **44** from *T. cuspidata* at the IC_50_ values of 16.88 μg/mL and 20.20 μg/mL, respectively, showed higher antioxidant activity because of the DPPH radical clearance rate compared to the IC_50_ value of 14.48 μg/mL for Vitamin C, which may be relevant to both compounds’ benzene ring polyhydroxyl groups for the effective scavenging of free radicals [51]. The concentration of 0.5 to 32.0 μg/mL in each flavonoid of apigenin **1**, luteolin **2**, kaempferol **18**, and quercetin **23** considerably displayed antioxidative effects according to DPPH-, ABTS radical-, and ferric-reducing experiments [9]. The capacity of hydroxyl radical scavenging was reported from kaempferol-3-o-rutinoside **19** at IC_50_ of 351.46 ± 2.30 μg/mL, and the suppression of a hyaluronidase at IC_50_ of 84.07 ± 10.46 μg/mL was seen as a powerful antiaging activity [43].

#### 3.4.3. Anti-Alzheimer’s Activities

Sciadopitysin **5** isolated from a 95% ethanol extract of *T. chinensis* was found to exhibit an inhibitory effect on amyloid beta (Aβ) peptide aggregation and on the formation of fibrils with anti-AD (Alzheimer’s disease) activity without toxicity to primary cortical neurons. The latter’s cellular test revealed its promotion effect on the proliferation of the human neuroblastoma cell line SH-SY5Y and indicated neuroprotection properties for the damage of primary cortical neurons motivated by Aβ protein. It can be a novel compound for potential therapeutics in AD [99].

#### 3.4.4. Antidiabetes Activities

The process of nonenzymatic glycation included the reduction of sugars and proteins through carbonyl groups’ reaction in vivo that was seen as a vital factor in the course of metabolic and pathophysiological processes leading to diabetes [100]. Diabetes-related complications were reported to be relevant to protein glycation induced by methylglyoxal as an important pathological factor. A reported flavonoid compound fisetin, as a flavonol, was found to significantly reduce kidney hypertrophy and albuminuria in diabetic mice model predominantly by decreasing the progress of the aforesaid glycation [101].

*T. chinensis* leaf tea demonstrated particularly strong antiglycative activity only in a 50 μg/mL concentration, and its mechanism has been proved through scavenging methylglyoxal in glucose-bovine serum albumin and fructose-bovine serum albumin models as well as human umbilical vein endothelial cell (HUVEC) models. Then its bioactive compounds of inhibiting glycation were isolated and preliminarily confirmed as (+)-catechin **43**, (-)-epicatechin **44**, gallocatechin **45**, epigallocatechin **46**, and procyanidin B2 **49** [61]. However, (+)-catechin **43** and (-)-epicatechin **44** from *T. cuspidata* with IC_50_ at 0.752 mg/mL and 0.655 mg/mL, respectively, demonstrated another mechanism, namely, they showed an obvious inhibitory effect on α-amylase [51], which may be relevant as a reason for existence of the 3-OH substituent group; in addition, the hydroxyl groups on the B rings in catechins and epicatechins are beneficial for the combination of the compound and enzyme, therefore expanding their effect of inhibiting α-amylase [102].

Based on the mechanism of upregulating nuclear factor erythroid 2 (Nrf2), procyanidin B2 **49** protected the endothelial progenitor cells (EPCs) function, lowered oxidative damage, and facilitated diabetic wound repair and angiogenesis in diabetic mice [65]. Quercetin-3-O-α-L-arabinopyranosyl-(1′′′→6″)-β-D-glucopyranoside **27** elevated glucose uptake and glycogen synthesis at 20 μmol/L through the IRS-1/PI3K/Akt/GSK-3β pathway [50]. The improvement of cell viability and superoxide dismutase activity, and reduction of reactive oxygen species generation, was conducted by eriodictyol **32** at 5, 10, and 20 μmol/L in diabetic mice [103]. The exacerbated course of diabetic disease aroused brain-damaging consequences such as memory loss, and it could be intervened by butin **33** at 10, 20 mg/kg through significantly abating the level of blood glucose, oxidative stress and neuroinflammation, and heightening the neurobehavioral parameters and metabolic levels in a most reptozotocin-treated model [55]. A triggered enzymatic hydrolysis by α-glucosidase, α-amylase made starch produce monosaccharides. Taxifolin **37,** as a competitive inhibitor, inhibited both enzymes at IC_50_ of 0.038, 0.647 mg/mL probably through the effect of hydrogen bond, π-π stack. Postprandial hyperglycemia was also regulated dramatically by taxifolin [58].

#### 3.4.5. Anticancer Activities

Three flavonoids including (+)-catechin **43**, (-)-epicatechin **44,** and quercetin-3-O-glucoside **29** from *T. cuspidata* were identified and proved to have significant antitumor effects on three cancer cells of MCF-7, Hela, and HepG2. Compound **29** showed the best anticancer effect against MCF-7 and Hela cells with IC_50_ at 36.4 μmol/L and 52.5 μmol/L, in consequence [51]. High doses of radiation are the vital reason for severe side effects in cancer radiation therapy. (-)-Epicatechin **44** decreased radiation resistance and improved the therapy effects through triggering coxidase (COX) activity in pancreatic cancer cells at concentrations of up to 200μmol/L [62]. An amount of 50 μmol/L of (-)-epicatechin **44** was proved to play a radioprotective role in human fibroblasts, and low concentration of 20 μmol/L also exhibited the same effects without solely abating clonogenic survival of human normal fibroblasts cells [63]. Similarly, quercetin 3-O-rutinoside **24** had a radioprotective effect on intestinal cancer through regulating ROS levels and antioxidative proteins and inhibiting the activation of inflammasome at 10, 25mg/kg [46].

A P-glycoprotein (P-gp) efflux transporter was seen as a major obstacle to the intestinal uptake of paclitaxel due to its poor aqueous solubility, followed by quick transportation by P-gp. Cytochrome P450 3A4 (CYP3A4) in the liver mainly induced the modulation of paclitaxel metabolism. Five biflavones identified as bilobetin **3**, sciadopitysin **5**, ginkgetin **6**, amentoflavone **7**, and sequoiaflavone **8**, from *T. yunnanensis*, as inhibitors of P-gp and CYP3A4, could elevate the oral absorption of paclitaxel through restraining P-gp activity at a concentration of 50 mg/mL; concurrently, they abated the expression and activity of CYP3A4 at a concentration of 100 μg/mL [104].

Apigenin **1** had a synergistic effect along with paclitaxel in cervical carcinoma (HeLa) cells, leading to 29% decrease of cell viability and 24% improvement of cell apoptosis with the combination index of 0.3918 ± 0.0436 [105]. Similarly, there was a synergistic inhibitory effect on human colorectal carcinoma resulting from luteolin **2** and oxaliplatin [10].

Human breast carcinoma (MCF-7) cells were suppressed dramatically by 4′′′-O-Methylamentoflavone **4** with ED_50_ of 4.56–16.24 μg/mL, further bringing cell apoptosis [15]. Likewise, ginkgetin **6** showed the inhibition effect on hepatocellular cancer cell line (HepG2) cells at 50.0 μmol/mL, leading to the decline of cell viability, cell numbers, and morphological changes [29]. Isoginkgetin **9** at from 2.5 to 20 μmol/L was reported to have an apparent inhibitory effect on A549 lung cancer cells via upregulating the expression of miR-27a-5p and downregulating the level of apurinic/apyrimidinic endo-deoxyribonuclease 1 (APEX1) [34], and sotetsuflavone **11** at 200 mmol/L showed the same inhibition of A549 cells but with relation to increased E-cadherin and decreased N-cadherin [37]. Loaded Naringenin **34** as nanoparticles also decreased the proliferation and migration of A549 cells [56].

#### 3.4.6. Antidepressant Activities

Apigenin **1** demonstrated the antidepressant activity according to both model tests, manifesting increased time of immobilization, swimming, and climbing tests in mice at 50 mg/kg. Further investigation could prove the mechanism through the activation of relative receptors of epinephrine, dopamine, and 5-HT3 [11].

#### 3.4.7. Neuronal Protective Activities

Neuronal cells can be damaged by diabetic glycation, producing neurodegenerative disorders such as Alzheimer’s disease. The viability of SK-N-MC neuronal cells was heightened apparently by sciadopitysin **5** at 400 μmol/mL, and cell apoptosis was inhibited in part at 0.1–1 μmol/L [27]. There was a similar result in which aromadendrin **38** enhanced the viability at 20 µmol/L and expanded the confluency at 2 mmol/L in SH-SY5y cells [59].

#### 3.4.8. Antileishmaniasis Activities

Leishmaniasis, as a disease of protozoan parasites, can disseminate from sand fly to human but has no vaccination prevention. Although modern drugs such as amphotericin B show certain therapeutic effects, they still have the problems of drug resistance and side effects [106]. Amentoflavone **7** had apparent inhibition effect on *Leishmania amazonensis* at IC_50_ of 28.5 ± 2.0 μmol/L through destroying the mitochondrial structure [30].

#### 3.4.9. Anti-Inflammatory, Antinociceptive, and Antiallergic Activities

Apigenin **1**, luteolin **2**, kaempferol **18**, and quercetin **23** proved to reduce the content of NO and phagocytosis from 50 to 200 μmol/L in an anti-inflammatory evaluation [9]. Putraflavone **10** and podocarpus flavone A **4** showed anti-inflammatory effects through inhibiting reactive oxygen species and the CD69 level [35]. Naringenin **34** was loaded as nanoparticles at 10 nmol/L, which brought the effect of attenuating the cytokines and their expression levels, containing IL-1β, IL-8, TNF-α, and IL-6 [56]. Butein **57** had significant anti-inflammatory and antinociceptive effects based on the decrease of nociception in the thermal and paw edema experiments at 10 to 20 mg/kg in mice, with depletion of inflammatory cytokines levels including TNF-α, IL-1β and IL-6 [69].

An anti-inflammatory and antiallergic activities evaluation was conducted with quercetin **23**, naringenin **34**, and naringenin chalcone **59.** There was a significant decrease of ear swelling derived from compounds **34** and **59** in ear edemas model. Four compounds showed apparent antiallergic effects probably through inhibiting the release of mast cells. As for the IgE-mediated passive cutaneous anaphylaxis experiment, all compounds inhibited allergic reaction via intravenous administration [45].

#### 3.4.10. Antivirus Activities

TheCOVID-19 pandemic is an ongoing coronavirus disease brought by severe acute respiratory syndrome coronavirus 2 (SARS-CoV-2) [107]. Kayaflavone **12** and amentoflavone **7** were speculated with molecular docking for their inhibitory effect on SARS-CoV-2. The results demonstrated that both compounds could combine with residues lining the catalytic site in the virus, and methyl transferase through hydrogen bonds overlapped with the ring of the S-adenosylmethionine [31]. Gallocatechin **45** proved to exhibit a suppression effect on SARS-CoV-2 at IC_50_ of 13.14 ± 2.081 μmol/L. There was the bind combination of π-π stacking and hydrogen bonds between the compound and virus [64]. Isorhamnetin **26** combined with the SARS-CoV-2 receptor, human angiotensin-converting enzyme 2, suppressed virus growth and invasion of the human body [48].

In addition, some other flavonoids also showed antivirus activities. In the early infection phase, kaempferol **18** and kaempferol-7-O-glucoside **20** at 100 μg/mL were proved to have potent antivirus activities through inhibiting HIV-1 reverse transcriptase. Compound **20** showed higher inhibitory effect than compound **18 [41]**. Myricetin **21** depicted an apparent restrained effect on infectious bronchitis virus (IBV)at 100 μmol/mL, attenuating the IBV activity closely 50% at 10 μmol/mL by regulating nuclear factor kappa-light-chain-enhancer of activated B cells (NF-κB) and interferon regulatory factor 7 (IRF7) pathways [44]. Different influenza virus evaluations were performed using Quercetin-7-O-glucoside **25**, and the result showed there was a strong inhibitory effect on virus strains with IC_50_ of 3.1 to 8.19 µg/mL, accompanied with the decrease of influenza-induced reactive oxygen species and autophagy [47]. Zika virus (ZIKV) transmission originates from the bite of female mosquitoes and leads to the fever and microcephaly or brain malformations in neonates. Pinocembrin **31** presented the significant suppression of ZIKV invasion at IC_50_ of 17.4 μmol/L via curbing virus RNA and protein [54].

#### 3.4.11. Antilipase Activities

Obesity is a complex medical condition that increases the risk of metabolic diseases, cardiovascular disease, depression, and cancer. Pancreatic lipase closely linked to the metabolism of triglycerides as an obesity factor [108]. Quercetin 3-rhamnoside **30** clearly showed inhibitory activity of lipase from 0 to 3 × 10^−5^ mol/L through the combination of the compound and some amino acids of lipase [52].

#### 3.4.12. Promotion of Melanogenesis

Melanogenesis disorders lead to some diseases, such as vitiligo. Tyrosinaseis a special enzyme regulating the rate-limiting reactions that produce melanin biosynthesis. Pinostrobin **35** inhibited tyrosinase activity at IC_50_ of 700 μmol/L, which further revealed the existence of non-covalent interactions and hydrogen bonds during the course of intermolecular binding by molecular docking [57].

#### 3.4.13. Hepatic-Protective Activities

The liver, as an important organ of drug detoxification, is easy to damage with many drugs such as Doxorubicin. Isoliquiritigenin **56** demonstrated significant hepatic-protective activity at 10 µmol/L, attenuated the levels of transaminases and inflammation cytokines, and improved catalase level through affecting the path of silent information regulator 1 [68].

## 4. Conclusions

The medicinal value of the *Taxus* genus has attracted worldwide attention due to taxanes having significant anticancer effects, and many other studies are in process. In this instance, the total of 59 flavonoids found in *Taxus* plants contain different skeletons, including flavones, biflavones, flavonols, flavonol glycosides, dihydroflavones, dihydroflavonols, dihydroflavonol glycoside, flavanols, biflavanols, and chalcones. Biflavones, chalcones, dihydroflavones, and flavanols have been found to accumulate in some specific *Taxus* plants, such as *T. baccata*, *T. celebica*, *T. media*, *T. cuspidata*, *T. mairei*, and *T. yunnanensis*. Biflavones and non-dimeric flavonoids exist with a rule of opposite distribution, indicating a relationship of ebb and flow.

In addition, biological activities of the flavonoids in *Taxus* plants have been proved including antibacterial, antiaging, anti-Alzheimer’s, antidiabetes, anticancer, antidepressant, antileishmaniasis, anti-inflammatory, antinociceptive and antiallergic, antivirus, antilipase, neuronal protective, and hepatic-protective activities, as well as the promotion of melanogenesis. Several flavonoids can ameliorate oral absorption of paclitaxel. Interestingly, the structure-activity relationships of flavonoids have been proved to derive from some special groups, such as the methoxyl group for the elevation of antifungal activity, two ortho hydroxy groups or the benzene ring polyhydroxyl group for antioxidant activities, and the 3-OH substituent group and the hydroxyl groups on B rings for antidiabetic activities, which indicates some aspects to be further studied and explored. Specially, some novel flavonoids have been found to show therapeutic potential for AD, as well as anticancer activities toward MCF-7, Hela, and HepG2 cells. The improvement in oral absorption of paclitaxel is found by some flavonoids through inhibiting P-gp activity and downregulating the expression and activity of CYP3A4.

In this review, the research result of chemical and pharmacological characteristics of flavonoids from the *Taxus* species could promote better use of these plants. However, this remains in the chemical experimental stage of study or cell investigation. Thus, a deeper investigation of pharmacological effects and mechanisms should be performed and verified to promote research and the application of *Taxus* flavonoids.

## Figures and Tables

**Figure 1 molecules-28-01713-f001:**
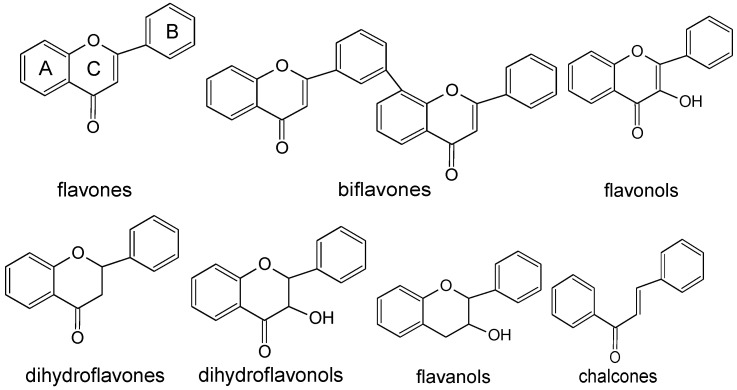
The structures of flavonoids from *Taxus* plants.

**Figure 2 molecules-28-01713-f002:**
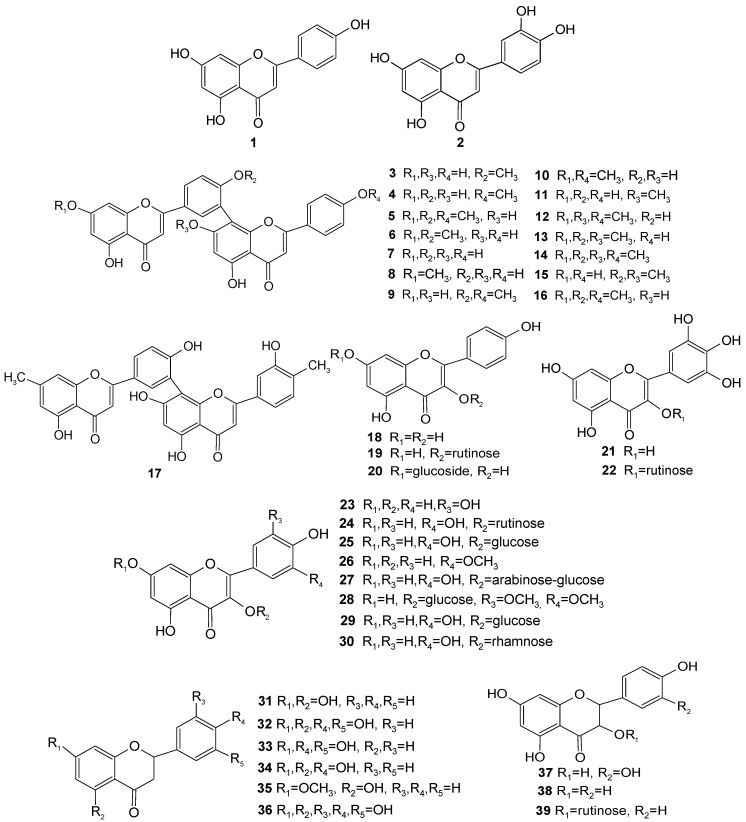
Chemical structures of flavonoids (**1**–**59**) in *Taxus* plants.

**Table 1 molecules-28-01713-t001:** Flavonoid compounds, distribution, and bioactivities from the *Taxus* genus.

No.	Type	Compound	Molecular Formula	Molecular Weight(Da)	Species (Part)	Bioactivity
1	Flavone	Apigenin	C_15_H_10_O_5_	270.24	*T. fuana, T. yunnanensis* (twigs); *T. baccata* (needles) [7,8]	Antioxidant, anticancer, antidepressant, anti-inflammatory activities [9,10,11]
2	Flavone	Luteolin	C_15_H_10_O_6_	286.24	*T. fuana, T. yunnanensis* (twigs) [7]	Antioxidant, anticancer, anti-inflammatory activities [9,10]
3	Biflavone	Bilobetin	C_31_H_20_O_10_	552.48	*T. baccata, T. celebica* (needles); *T. chinensis, T. cuspidata, T. media* (twigs, leaves) [12,13]	Antibacterial activities [12]
4	Biflavone	4′′′-O-Methylamentoflavone(or podocarpusflavone-A)	C_31_H_20_O_10_	552.48	*T. baccata*, *T. media* (needles) [9,14]	Anticancer activities [15]
5	Biflavone	Sciadopitysin	C_33_H_24_O_10_	580.54	*T. baccata, T. media, T. celebica* (needles, leaves); *T. cuspidata* (twigs, bark, leaves, branches); *T. brevifolia* (leaves); *T. canadensis* (leaves, twigs); *T. wallichiana* (leaves); *T. mairei* (leaves); *T. chinensis, T.media* (twigs, leaves); *T. cuspidate var. nana* (leaves) [4,12,13,16,17,18,19,20,21,22,23,24,25,26]	Anti-Alzheimer’s disease, antibacterial, neuronal protective activities [12,27]
6	Biflavone	Ginkgetin	C_32_H_22_O_10_	566.51	*T. baccata* (needles, leaves), *T. cuspidata* (twigs, bark, leaves, branches); *T. canadensis* (leaves, twigs); *T. chinensis var. mairei, T. media* (fruits); *T. chinensis, T. media* (twigs, leaves, fruits); *T. wallichiana* (leaves); *T. fuana, T. yunnanensis* (twigs); *T. cuspidate var. nana* (leaves) [4,7,12,13,16,17,18,19,20,22,23,24,26,28]	Antibacterial, anticancer activities [12,29]
7	Biflavone	Amentoflavone	C_30_H_18_O_10_	538.46	*T. baccata* (needles); *T. wallichiana*(leaves); *T. fuana, T. yunnanensis* (twigs) [7,12,23,24]	Antibacterial, antileishmaniasis, antivirus activities [12,30,31]
8	Biflavone	Sequoiaflavone	C_31_H_20_O_10_	552.48	*T. baccata* (needles, leaves); *T. media* (needles); *T. wallichiana* (leaves); *T. canadensis* (needles); *T. mairei, T. chinensis* (leaves) [4,12,14,16,23,24,25,32]	Antibacterial activities [12]
9	Biflavone	Isoginkgetin	C_32_H_22_O_10_	566.51	*T. chinensis, T. cuspidata, T. media* (twigs, leaves); *T. chinensis var. mairei* (twigs) [13,33]	Anti-inflammatory activities [34]
10	Biflavone	Putraflavone	C_32_H_22_O_10_	566.51	*T. canadensis* (needles); *T. chinensis var. mairei* (twigs) [32,33]	Anti-inflammatory activities [35]
11	Biflavone	Sotetsuflavone	C_31_H_20_O_10_	552.48	*T. baccata* (not mentioned); *T. cuspidata* (leaves) [19,36]	Anticancer activities [37]
12	Biflavone	Kayaflavone	C_33_H_24_O_10_	580.54	*T. cuspidata* (leaves); *T. baccata* (needles) [17,19,36]	Antivirus activities [31]
13	Biflavone	4′,7,7″-Tri-O-methyl amentoflavone	C_33_H_24_O_10_	580.54	*T. baccata* (leaves) [36,38]	Not reported
14	Biflavone	4′,4″,7,7″-Tetra-O-methyl amentoflavone	C_34_H_26_O_10_	594.56	*T. baccata* (needles, leaves) [36,38]	Not reported
15	Biflavone	4′,7″-Di-O-methyl amentoflavone	C_32_H_22_O_10_	566.51	*T. baccata* (needles) [36]	Not reported
16	Biflavone	4″-O-methyl ginkgetin	C_33_H_24_O_10_	580.54	*T. chinensis var. mairei, T. media* (fruits) [28]	Not reported
17	Biflavone	3″-hydroxy-4″,7-dimethyl amentoflavone	C_32_H_22_O_9_	550.51	*T. canadensis* (needles) [39]	Not reported
18	Flavonol	Kaempferol	C_15_H_10_O_6_	286.24	*T. brevifolia* (leaves); *T. baccata* (needles); *T. fuana, T. yunnanensis* (twigs); *T. mairei* (twigs) [7,14,21,40]	Antioxidant, antivirus, anti-inflammatory activities [9,41]
19	Flavonol glycoside	Kaempferol-3-O-rutinoside	C_27_H_30_O_15_	594.52	*T. baccata* (needles, twigs); *T. chinensis var. mairei* (twigs) [14,33,42]	Antioxidant activities [43]
20	Flavonol glycoside	Kaempferol-7-O-glucoside	C_21_H_20_O_11_	448.38	*T. baccata* (needles); *T. chinensis var. mairei* (twigs) [14,33]	Antivirus activities [41]
21	Flavonol	Myricetin	C_15_H_10_O_8_	318.23	*T. baccata* (needles) [14]	Antivirus activities [44]
22	Flavonol glycoside	Myricetin-3-O-rutinoside	C_27_H_30_O_17_	626.52	*T. baccata* (needles) [14]	Not reported
23	Flavonol	Quercetin	C_15_H_10_O_7_	302.24	*T. brevifolia* (leaves); *T. cuspidate* (bark, leaves); *T. baccata* (needles, twigs); *T. fuana, T. yunnanensis* (twigs); *T. chinensis*, *T. cuspidata, T. media* (twigs, leaves); *T. mairei* (twigs); *T. chinensis var. mairei* (twigs); *T. cuspidate var. nana* (leaves) [7,13,14,18,21,26,33,40,42]	Antioxidant, anti-inflammatory, antiallergic activities [9,45]
24	Flavonol glycoside	Quercetin-3-O-rutinoside (or rutin)	C_27_H_30_O_16_	610.52	*T. baccata* (needles or leaves, twigs); *T. chinensis var. mairei* (twigs) [14,33,42]	Anticancer activities [46]
25	Flavonol glycoside	Quercetin-7-O-glucoside	C_21_H_20_O_12_	464.38	*T. baccata* (needles) [14]	Antivirus activities [47]
26	Flavonol	Isorhamnetin	C_16_H_12_O_7_	316.26	*T. brevifolia* (leaves); *T. cuspidate* (bark, leaves); *T. cuspidate var. nana* (leaves); *T. baccata* (needles) [8,18,21,26]	Antivirus activities [48]
27	Flavonol glycoside	Quercetin-3-O-α-L-arabinopyranosyl-(1′′′→6”)-β-D-glucopyranoside	C_26_H_28_O_16_	596.49	*T. cuspidata* (needles) [49]	Antidiabetes activities [50]
28	Flavonol glycoside	Tricin-3-O-glucoside	C_23_H_24_O_13_	508.43	*T. chinensis var. mairei, T. media* (fruits) [28]	Not reported
29	Flavonol glycoside	Quercetin-3-O-glucoside	C_21_H_20_O_12_	464.38	*T. cuspidata* (branches, leaves); *T. chinensis*, *T. cuspidata, T. media* (twigs, leaves) [13,51]	Anticancer activities [51]
30	Flavonol glycoside	Quercetin 3-rhamnoside	C_21_H_20_O_11_	448.38	*T. chinensis, T. cuspidata, T. media* (twigs, leaves) [13]	Antilipase activities [52]
31	Dihydroflavone	Pinocembrin	C_15_H_12_O_4_	256.25	*T. mairei* (twigs) [53]	Antivirus activities [54]
32	Dihydroflavone	Eriodictyol	C_15_H_12_O_6_	288.25	*T. mairei* (twigs) [53]	Antidiabetes activities [55]
33	Dihydroflavone	Butin	C_15_H_12_O_5_	272.25	*T. mairei* (twigs) [53]	Antidiabetes activities [55]
34	Dihydroflavone	Naringenin	C_15_H_12_O_5_	272.25	*T. media, T. cuspidata* (twigs); *T. chinensis var. mairei* (twigs) [33,53]	Anticancer, anti-inflammatory, antiallergic activities [45,56]
35	Dihydroflavone	Pinostrobin	C_16_H_14_O_4_	270.28	*T. media, T. cuspidata* (twigs) [53]	Promotion of melanogenesis [57]
36	Dihydroflavone	Dihydrotricetin	C_15_H_12_O_7_	304.25	*T. media, T. cuspidata* (twigs) [53]	Not reported
37	Dihydroflavonol	Taxifolin	C_15_H_12_O_7_	304.25	*T. baccata* (needles) [8]	Antidiabetes activities [58]
38	Dihydroflavonol	Aromadendrin	C_15_H_12_O_6_	288.25	*T. chinensis var. mairei, T. media* (fruits) [28]	Neuronal protective activities [59]
39	Dihydroflavonol glycoside	Aromadendrin-3-O-rutinoside	C_27_H_32_O_15_	596.53	*T. chinensis var. mairei, T. media* (fruits) [28]	Not reported
40	Flavanol	5-deoxyleucopelargonidin	C_15_H_14_O_5_	274.27	*T. media* (twigs) [53]	Not reported
41	Flavanol	Leucopelargonidin	C_15_H_14_O_6_	290.27	*T. media* (twigs); *T. chinensis var. mairei, T. media* (fruits) [24,53]	Not reported
42	Flavanol	Leucocyanidin	C_15_H_14_O_7_	306.27	*T. media* (twigs) [53]	Not reported
43	Flavanol	(+)-Catechin	C_15_H_14_O_6_	290.27	*T. cuspidata* (needles, wood, roots); *T. fuana, T. yunnanensis* (twigs); *T. chinensis* (leaves) [7,49,60,61]	Antioxidant, antidiabetes, anticanceractivities [3,51,60,61]
44	Flavanol	(-)-Epicatechin	C_15_H_14_O_6_	290.27	*T. cuspidata* (needles, wood, roots); *T. fuana, T. yunnanensis* (twigs); *T. chinensis* (leaves) [7,49,60,61]	Antioxidant, antidiabetes, anticanceractivities [51,60,61,62,63]
45	Flavanol	Gallocatechin	C_15_H_14_O_7_	306.27	*T. chinensis* (leaves) [61]	Antidiabetes, antivirus activities [61,64]
46	Flavanol	Epigallocatechin	C_15_H_14_O_7_	306.27	*T. chinensis* (leaves) [61]	Antidiabetes activities [61]
47	Flavanol	(+)-Catechin pentaacetate	C_25_H_24_O_11_	500.45	*T. mairei* (twigs) [40]	Not reported
48	Flavanol	(-)-Epicatechin pentaacetate	C_25_H_24_O_11_	500.45	*T. mairei* (twigs) [40]	Not reported
49	Flavanol	Procyanidin B2	C_30_H_26_O_12_	578.52	*T. chinensis* (leaves) [61]	Antidiabetes activities [51,65]
50	Biflavanol	Procyanidin B-2 decaacetate	C_50_H_46_O_22_	998.89	*T. mairei* (twigs) [40]	Not reported
51	Biflavanol	Procyanidin B-3-decaacetate	C_50_H_46_O_22_	998.89	*T. mairei* (twigs) [40]	Not reported
52	Biflavanol	Procyanidin B-4-decaacetate	C_50_H_46_O_22_	998.89	*T. mairei* (twigs) [40]	Not reported
53	Biflavanol	Afzelechin-(4α→8)-afzelechin	C_30_H_26_O_10_	546.52	*T. cuspidata* (roots) [66]	Not reported
54	Biflavanol	Afzelechin-(4α→8)-afzelechin octaacetate	C_46_H_42_O_18_	882.81	*T. cuspidata* (roots) [66]	Not reported
55	Chalcone	Pinocembrin chalcone(or 2′,4′,6′-trihydroxychalcone)	C_15_H_12_O_4_	256.25	*T. mairei* (twigs) [53]	Antibacterial activities [67]
56	Chalcone	Isoliquiritigenin(or 2′,4′,4′-trihydroxy chalcone)	C_15_H_12_O_4_	256.25	*T. media* (twigs) [53]	Hepatic-protective activities [68]
57	Chalcone	Butein	C_15_H_12_O_5_	272.25	*T. media* (twigs) [53]	Anti-inflammatory, antinociceptive activities [69]
58	Chalcone	Homoeriodictyol chalcone	C_16_H_14_O_6_	302.28	*T. media* (twigs) [53]	Not reported
59	Chalcone	Naringenin chalcone	C_15_H_12_O_5_	272.25	*T. media, T. cuspidata* (twigs) [53]	Anti-inflammatory activities [45]

## Data Availability

Not applicable.

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
