# Peer review of "Flavonoid Components, Distribution, and Biological Activities in *Taxus*: A review"

_molecules, 2023, doi:10.3390/molecules28041713_

Round 1
Reviewer 1 Report
Manuscript ID: molecules-2179780
Type of manuscript: Review
Title: Flavonoids Components, Distribution and Biological Activities in Taxus:A review
Authors: Qiang Wei *, Zhao Qi Li, Lin Rui Wang Submitted to section:
Computational and Theoretical Chemistry,
Journal: Molecules
The manuscript can be accepted after correcting the following mistakes.
1. Lines 8 and 9: Taxol and Docetaxel proper names are capitalized - to be corrected
2. Line 31-32: What is A, B C? Draw a general formula or mark on Fig. 1
3. Line 42 and 53: backbone replaced with a skeleton
4. Figure 1 lack of glycosides (line 38 …as seen in Figure 1)
5. Line 48:…. distinct name from flavonols (with” a” )? - explain or give the name
6. Line 49-52: obvious school knowledge why write this
7. Line 55-58: is this the aim of the work?
8. Line 62: species should be listed
9. Line 62 .. genus….add the name
10. Table 1. Molecular weight has more exact values (e.g. 270.24) and units. In table -No compounds, Name of compounds, parts of what? Maybe add a pharmacological activity?
11. Improve Table 1 graphically in general
12. No description under Table 1 to Figure 2. What is there and why etc?
13. Figure 2 put in order : 3-6… R1,R3,R4=H, R2=H etc (for all structures)
14. Figure 2 compound 23 error R3=OH. This should be corrected
15. Section 2.2 line 91: divide (1), (2), (3) with a space in the text marking the manuscript. The same is in chapter 3.
16. Line 205: positive effects …… it should be corrected
17. Line 209- 210: Kiessler ?, Berk.Curt?.....
18. Line: 215-219
19. description of research chaotic. List which 4 compounds. How were they studied (who studied them)? What were the reference drugs?
20. Line 247: in vivo- italics
21. Section 4.5 lacks reference drugs used in the study
22. Section 4.6 should be a subsection of section 4.5
23. Line 277: five biflavones … as bilobetin 3 insert compound numbers
24. throughout the text, remove parentheses next to compound numbers (or throughout the manuscript, use them)
25. Line 293-294: …and impovement of…- II separate sentence
26. Lines 292-294 and 299-302 are about the same thing.
27. Line 306-309:? The sentence makes no sense and should be corrected.
28. References correct according to editorial guidelines
1-font
Missing pages 8,17,21,20,45,48,42
29. Minor mistakes in language throughout the manuscript e.g:
Line28: other
Line 31 a C6-….
Line: 34,133,263,275,164,194…. –lack of the
Line 36: found in the plants
Line 52: flavans
Line 82: an additional
Line 83: compared
Line 136: falvanols
Line 115: compounds
Line 161 in Table
Line 292: the biological activities…
etc…
Author Response
Dear Reviewer:
Thank you for your revised suggestions, which gave me a chance to improve the manuscript quality. I am so sorry for a little late reply to the revised manuscript. The details can be seen in the attachment. Thanks a lot.
Reviewer 1
The manuscript can be accepted after correcting the following mistakes.
- Lines 8 and 9: Taxol and Docetaxel proper names are capitalized - to be corrected
Answer: It was corrected in the abstract.
- Line 31-32: What is A, B C? Draw a general formula or mark on Fig. 1
Answer: It was corrected on Fig. 1.
- Line 42 and 53: backbone replaced with a skeleton
Answer: It was corrected.
- Figure 1 lack of glycosides (line 38 …as seen in Figure 1)
Answer: To keep the accordance with the words. Line 38 ....delete the "the glycosides"
- Line 48:…. distinct name from flavonols (with” a” )? - explain or give the name
Answer: explain as follows: They have a distinct name from flavanols (flavonols' first "o" is replaced)
- Line 49-52: obvious school knowledge why write this
Answer: It is simplified now.
- Line 55-58: is this the aim of the work?
Answer: It was corrected
- Line 62: species should be listed
Answer: It was corrected.
- Line 62 .. genus….add the name
Answer: It was corrected.
- Table 1. Molecular weight has more exact values (e.g. 270.24) and units. In table -No compounds, Name of compounds, parts of what? Maybe add a pharmacological activity?
Answer: It was corrected.
- Improve Table 1 graphically in general
Answer: It was corrected.
- No description under Table 1 to Figure 2. What is there and why etc?
Answer: They are explained separately in 3.1. Chemical Constituents and 3.2.Flavonoid distribution 5. Flavonoids's ioactivities
- Figure 2 put in order : 3-6… R1,R3,R4=H, R2=H etc (for all structures)
Answer: It was corrected.
- Figure 2 compound 23 error R3=OH. This should be corrected
Answer: It was corrected.
- Section 2.2 line 91: divide (1), (2), (3) with a space in the text marking the manuscript. The same is in chapter 3.
Answer: It was corrected.
- Line 205: positive effects …… it should be corrected
Answer: It was corrected.
- Line 209- 210: Kiessler ?, Berk.Curt?.....
Answer: It was corrected.
- Line: 215-219
Answer: It was corrected.
- description of research chaotic. List which 4 compounds. How were they studied (who studied them)? What were the reference drugs?
Answer: It was corrected.
- Line 247: in vivo- italics
Answer: It was corrected.
- Section 4.5 lacks reference drugs used in the study
Answer: It was corrected.
- Section 4.6 should be a subsection of section 4.5
Answer: It was corrected.
- Line 277: five biflavones … as bilobetin 3 insert compound numbers
Answer: It was corrected.
- throughout the text, remove parentheses next to compound numbers (or throughout the manuscript, use them)
Answer: It was corrected.
- Line 293-294: …and impovement of…- II separate sentence
Answer: It was corrected.
- Lines 292-294 and 299-302 are about the same thing.
Answer: It was corrected.
- Line 306-309:? The sentence makes no sense and should be corrected.
Answer: It was corrected.
- References correct according to editorial guidelines
1-font
Missing pages 8,17,21,20,45,48,42
Answer: It was corrected.
- Minor mistakes in language throughout the manuscript e.g:
Line28: other
Line 31 a C6-….
Line: 34,133,263,275,164,194…. –lack of the
Line 36: found in the plants
Line 52: flavans
Line 82: an additional
Line 83: compared
Line 136: falvanols
Line 115: compounds
Line 161 in Table
Line 292: the biological activities…
etc…
Answer: It was corrected.

Reviewer 2 Report
The subject of the manuscript is interesting but requires substantial improvements before publication. The Introduction section must be modified. There are no references to the purpose and novelty of this review. I suggest the introduction of a section related to the physico-chemical properties of flavonoids, maybe even methods of extraction, and isolation. The section on the biological activity of Taxus flavonoids should be significantly improved. In my opinion, a presentation of the biological activity/therapeutic action of each type of flavonoid compound would be useful. The Conclusions section should be revised.
Author Response
Dear Reviewer:
Thank you for your revised suggestions, which gave me a chance to improve the manuscript quality. I am so sorry for a little late reply to the revised manuscript. The details can be seen in the attachment. Thanks a lot.
Reviewer 2
The subject of the manuscript is interesting but requires substantial improvements before publication. The Introduction section must be modified. There are no references to the purpose and novelty of this review. I suggest the introduction of a section related to the physico-chemical properties of flavonoids, maybe even methods of extraction, and isolation. The section on the biological activity of Taxus flavonoids should be significantly improved. In my opinion, a presentation of the biological activity/therapeutic action of each type of flavonoid compound would be useful. The Conclusions section should be revised.
Answer:
1.physico-chemical properties , methods of extraction, and isolation has been added as seen 3.2 Flavonoid Properties, Extraction and Isolation
- In Table1, biological activities was added.
- The Introduction has added the purpose and novelty. " Nowadays, high and positive attention for health care has been focused on flavonoids. An European nutritional investigation proved that cancer had a direct relevance to flavonoids intake inpeople of different ethnicities[4]. Meanwhile, computer-aided workflows were used to design the drug based on flavonoid structures for better synthesis path, bioactive effects, and few side effects to the human body[5]"

Reviewer 3 Report
1/ There is a recently published book
Kumara Swamy, M., T. Pullaiah and Zhe-Sheng Chen. 2022. Paclitaxel: Sources, Chemistry, anticancer actions and Current Biotechnology. Elsevier.
This reference has not been quoted or cited anywhere in the paper.
2.Species name first letter should be small letter. Many names in references are in capital letter (upper cap). Please change them to lower cap (small letters)

Author Response
Dear Reviewer:
Thank you for your revised suggestions, which gave me a chance to improve the manuscript quality. I am so sorry for a little late reply to the revised manuscript. The details is can be seen in the attachment. Thanks a lot.
1/ There is a recently published book
Kumara Swamy, M., T. Pullaiah and Zhe-Sheng Chen. 2022. Paclitaxel: Sources, Chemistry, anticancer actions and Current Biotechnology. Elsevier.
This reference has not been quoted or cited anywhere in the paper.
Answer: This reference was added as reference 2.
2.Species name first letter should be small letter. Many names in references are in capital letter (upper cap). Please change them to lower cap (small letters)
Answer: It was corrected
Additionally, PDF part of the reviewer 3 was also corrected.
Other correction
1.English was edited through the correction of MDPI_english.
2.During the course of correction, there was some mistakes, so they are corrected simultaneously.

Round 2
Reviewer 2 Report
Agree with the revision form of the manuscript.